# Edge Grinding Characteristics of Display Glass Substrate

**Dennis Wee Keong Neo** [iD], **Kui Liu** *[iD], **Rui Huang and Hu Wu**

Singapore Institute of Manufacturing Technology, 73 Nayang Drive, Singapore 637662, Singapore;
dennis_neo@simtech.a-star.edu.sg (D.W.K.N.); huang_rui@simtech.a-star.edu.sg (R.H.);
hwu@simtech.a-star.edu.sg (H.W.)
* Correspondence: kliu@simtech.a-star.edu.sg; Tel.: +65-6793-8565

**Abstract:** Display glass substrate as a brittle material is very challenging to machine due to its excellent physical, mechanical, electrical, and optical properties such as high hardness, high strength, high wear resistance, good fracture toughness, good chemical stability, and good thermal stability. On the basis of Griffith fracture mechanics, our theoretical analysis indicated that edge grinding of the display glass substrate is under brittle mode when grinding with the given conditions, which was verified by the experimental studies of ground glass edge surface topography and fractured surface obtained. Grinding force ($F_y$) in the vertical direction was much larger than grinding force ($F_x$) in the horizontal direction, causing a large compressive stress acting on the grinding glass edge. Grinding torque was slightly increased with the increase of grinding speed. Grinding temperature was very high when measured under dry grinding compared with measurement under high-pressure coolant. Grinding of glass substrate edge was performed partially under ductile mode machining in the experimental conditions, which can be attributed to and contributed by those micro cutting edges generated by the fractured diamond grit on the grinding wheel surface.

**Keywords:** edge grinding; characteristics; glass substrate; display

## 1. Introduction

Thin-film transistor (TFT) liquid crystal displays (LCDs) and organic light emitting diode (OLED) displays are becoming the dominant form of flat displays used in many devices, especially portable ones such as smartphones, tablets, TVs, PCs, watches, digital cameras, digital camcorders, and game consoles, as well as instrument panels, aircraft cockpit displays, automotive navigation systems, and indoor and outdoor signage. This is because the device mass is largely reduced even with a very big size, and the device size can be very small while retaining a high-quality display. Due to the recent great growth of the market for portable consumer electronic devices and laptop computers, the use of high-quality flat displays such as LCDs and OLEDs has been increased dramatically [1]. Therefore, the market demand for display glass substrates has increased rapidly. Table 1 lists the panel size of glass substrates for display, showing that glass panels are becoming larger and larger over the years so as to improve their productivity [2]. Meanwhile, there is an increasing demand for further reducing the thickness of display glass substrate because of the market-driven of lighter devices and material saving.

Unfortunately, glass is very difficult to cut and is easily fractured during the machining process [3–5]. In as early as 1976, the first reproducible evidence of grinding ductility in glass workpiece was reported using a silicon carbide wheel exhibiting extensive plastic flow over the ground surface [6,7]. Lately, many works have been conducted to investigate the different aspects such as machining mechanism for diamond cutting of ZKN7 glass [8], cutting tool edge effect and material properties on micro/nano cutting [9], the brittle–ductile transition mechanism for ultrasonic vibration diamond cutting of glasses [10], ultraprecision ductile cutting of glass with ultrasonic vibration assistance to enhance their ductile mode cutting performance [11], analytically modeling of shear angle effect in

ultrasonic vibration-assisted micro-machining [12], and micromachining mechanics by considering chip thickness accumulation [13]. However, not much work has been done on grinding or/and machining of display glass substrates. The brittle nature of glass makes it very sensitive to tiny cracks, which can cause the display glass panels to break at stresses far below its theoretical strength [2].

**Table 1.** Panel size of different generations of display glass substrates.

| Generation | Year | Size (mm) |
|---|---|---|
| 1 | 1988 | 300 × 400 |
| 2 | 1993 | 360 × 465 |
| 3 | 1995 | 550 × 650 |
| 4 | 2000 | 730 × 920 |
| 5 | 2002 | 1100 × 1300 |
| 6 | 2003 | 1500 × 1850 |
| 7 | 2005 | 1870 × 2200 |
| 8 | 2008 | 2200 × 2500 |
| 9 | 2009 | 2400 × 2800 |
| 10 | 2010 | 2880 × 3130 |
| 10.5 | 2019 | 2940 × 3370 |

Display glass substrates are manufactured by the fusion process, where the molten glass is flowed over sides and drawn down to form a continuous flat piece of glass with very precise thickness controls, and then cut or scribed into the glass panel size. Edge cutting of display glass substrates is one of main sources for introducing cracks/defects and downgrading the material strength so that most glass panel failures initiate from an edge [2]. A vibration-assisted scribing technique of LCD glass substrate has been developed to automatically separate the glass substrate without the need of breaking process [14]. The formation of subsurface damage induced by grinding of LCD glass panels was investigated using two types of diamond wheels [15]. The chemo-mechanical grinding (CMG) process has been developed for the machining of quartz glass substrates to target a defect-free machining process [16]. More importantly, edge grinding is also one of the most efficient manufacturing processes for making display glass panels to remove the edge defects induced by the above-mentioned edge cutting process, of which the edge defects are replaced by multiple small flaws/cracks from the grinding process [15]. In this paper, edge grinding characteristics of display glass substrate will be systemically studied and analyzed in terms of theoretical analysis of ductile mode machining, grinding forces and torque, ground surface topography, and grinding wheel tool wear in order to understand the glass substrate grinding behavior and provide benefit to the industry and research communities.

## 2. Theoretical Analysis

When machining brittle materials, there is a brittle-to-ductile transition of machining modes when continuously reducing the cutting depth down to the micron and/or nanometer scale, which is largely depended on workpiece material's properties and its manufacturing processes [17]. According to the Griffith fracture mechanics [17,18], a critical depth-of-indentation or critical undeformed chip thickness at brittle-to-ductile transition region in machining of brittle materials can be predicted by

$$d_c = u \left( \frac{E}{H} \right) \left( \frac{K_C}{H} \right)^2 \tag{1}$$

where $d_c$ is the critical undeformed chip thickness or the critical depth of indentation, $H$ is material hardness, $E$ is Young's modulus, and $K_C$ is material fracture toughness. In addition, $u$ is the transition factor of the workpiece material from brittle mode machining to ductile mode machining, which is dominated by the workpiece material properties.

According to Equation (1), there is a critical value of undeformed chip thickness in machining of brittle materials, below which the chips will be formed under ductile mode machining. This critical value is a function of the brittle material's hardness, Young's modulus, and fracture toughness. The above theoretical description demonstrates that a critical depth of cut and an area of brittle-to-ductile transition exist during the machining of brittle materials, although the critical value may be varied and controlled by the workpiece inherent configuration and loading conditions.

The workpiece material used in this study is display glass substrate NA35, and its material properties are listed in Table 2. Substituting the values of glass's elastic modulus, hardness, and critical fracture toughness into Equation (1), as well as the value of transition factor from brittle mode to ductile mode $u$ of 0.15 [17], we determined the theoretical critical depth of cut $d_c$ for ductile mode machining of the display glass substrate as 40 nm.

**Table 2.** Material properties of glass substrate NA35.

| Material Properties | Display Glass |
|---|---|
| Specific gravity (g/cm)$^3$ | 2.49 |
| Young's modulus (GPa) | 70.9 |
| Modulus of rigidity | 28.3 |
| Modulus of volume elasticity | 45.2 |
| Knoop hardness (GPa) | 5.2 |
| Fracture toughness (MPa·m$^{0.5}$) | 0.7 |
| Poisson's ratio | 0.24 |
| Strain point (°C) | 661 |
| Annealing point (°C) | 715 |
| Softening point (°C) | 895 |
| Roughness (RMS, nm) $1 \times 1$ μm size | 0.324 |
| Refractive index (Nd) | 1.513 |
| Dielectric constant (S/m) | 4.7 |

Moreover, for grinding, the maximum undeformed chip thickness for an individual grinding abrasive grit, $h_{max}$, can be calculated by the following equation [19]:

$$h_{max} = \left[ \frac{3}{C tan\alpha} \left( \frac{v_w}{v_s} \right) \left( \frac{a_e}{d_s} \right)^{1/2} \right]^{1/2} \qquad (2)$$

where $C$ is the density of active grinding points, $\alpha$ is the semi-induced angle of cross section for the undeformed chip, $d_s$ is the grinding wheel diameter, $v_w$ is workpiece feed rate, $v_s$ is the grinding wheel speed, and $a_e$ is the wheel depth of cut. Thus, when grinding of NA35 glass with a given grinding condition, its maximum undeformed chip thickness $h_{max}$ can be calculated by Equation (2) accordingly.

### 3. Experiment Details

Edge grinding experiments of display glass substrates were carried out on an Okamoto CNC (computer numerical control) high-speed grinder using a diameter of 250 mm resin-bound diamond wheel with the diamond abrasive mesh size of 500 (grit size ranging from 31 μm to 37 μm) with a diamond abrasive concentration rate of 75%. The grinding wheel was pre-balanced using a Sigma dynamic balancer at the different grinding speeds to be employed in the late experiments. High pressure water-based coolant and grinding conditions used for the tests are listed in Table 3. Workpiece size was $300 \times 50 \times 0.7$ mm for the edge grinding tests. Figure 1 is a close view of the edge grinding experimental setup. A specially designed vise was used to clamp the display glass substrate with two pieces of rubber on each side of the clamping to protect the glass substrate.

**Table 3.** Edge grinding conditions used for glass substrates.

| Grinding Parameters | Grinding Conditions |
|---|---|
| Spindle speed (rpm) | 250, 500, 750, 1000, 2000, 3000 |
| Feed rate (mm/min) | 7000, 10,000 |
| Grinding depth (mm) | 0.1, 0.15, 0.2, 0.4 |
| High-pressure coolant | Water-based |

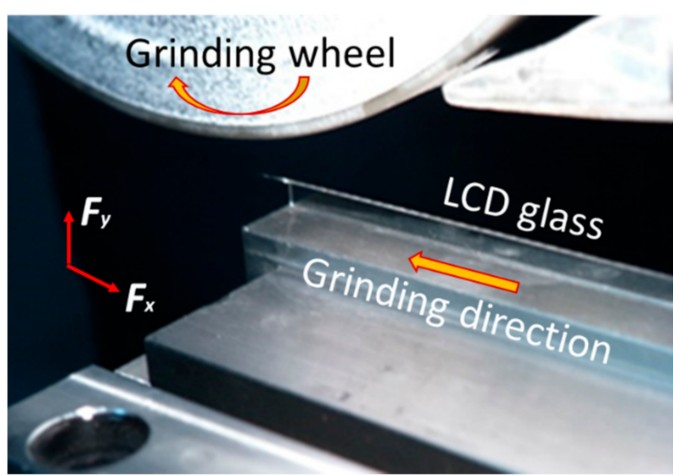

**Figure 1.** Close view of the edge grinding experimental setup.

Figure 2 shows the schematic diagram of the data acquisition system for edge grinding of display glass substrates, which was employed to collect the grinding forces and torque using 1 Kistler four-component rotary dynamometer and 1 Kistler three-component static dynamometer, as well as grinding wheel spindle current value using a current transducer. Grinding temperature was measured using a diameter of 0.5 mm thermocouple and a Pico recorder, as shown in Figure 3. Sand blasting was used to produce a half cylinder on a thickness of 0.7 mm glass substrate, and two pieces of such glass substrates were clamped together to form a blind hole so as to embed a diameter of 0.5 mm thermocouple inside, as shown in Figure 3a. Figure 3b shows the actual embedded thermocouples in glass substrates. Ground glass edge surface was examined using a Keyence optical microscope and a scanning electron microscope (SEM). Diamond grinding wheel wear was examined using an optical microscope and SEM.

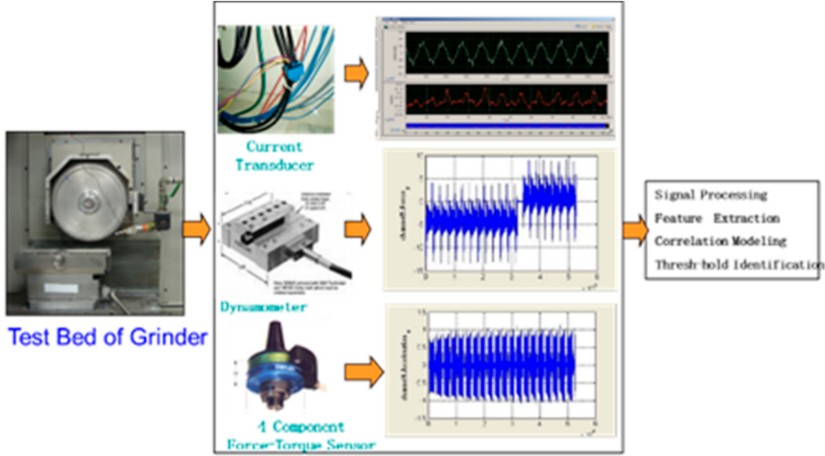

**Figure 2.** Data acquisition system for edge grinding of display glass substrates.

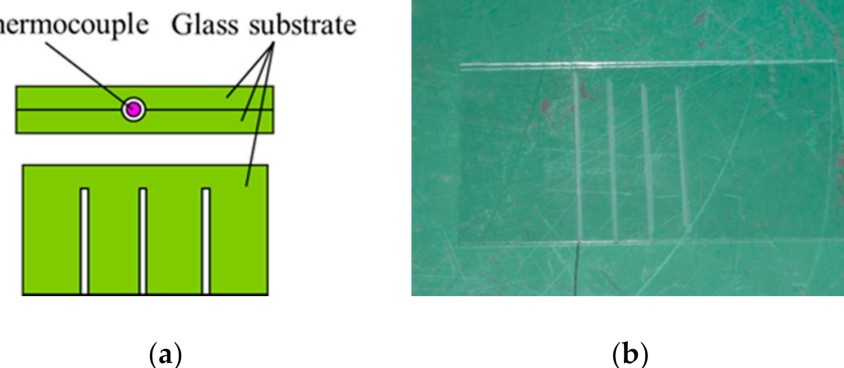

**Figure 3.** Experimental details of temperature measurement in edge grinding of glass substrates: (**a**) schematic illustration of embedded thermocouples; (**b**) actual embedded thermocouples in glass substrates.

## 4. Experimental Results

Edge grinding experiments were conducted and repeated to investigate its grinding performance through the plane grinding method using the diamond cut glass substrate NA35. Its edge grinding characteristics were examined and analyzed accordingly.

### 4.1. Grinding Force and Torque

Figure 4 shows an example of the measured grinding forces and spindle torque obtained in the plane grinding of glass substrate edges, where the grinding conditions used were grindings speed of 26.2 m/s (2000 rpm), feed rate of 10,000 mm/min, depth of cut of 0.15 mm, and high-pressure coolant. As the four-component Kistler dynamometer was mounted to the grinding spindle in the horizontal direction, which usually is mounted on a vertical milling spindle to measure machining forces and torque, the mass of the dynamometer together with the grinding wheel acted as a pre-loading. Thereafter, all the measured forces $F_x$ and $F_y$ had an initial value and the measured torque had an off-set value before the actual grinding in response to the pre-loading, as shown in Figure 4. As seen from Figure 4a,b, grinding force ($F_y$) in the vertical direction was about 10 times larger than grinding force ($F_x$) in the horizontal direction. It is clearly indicated that there was a very big compressive stress acting on the glass substrate workpiece, which was a potential source for glass substrate cracking and fracturing.

Figure 5 shows the effects of grinding distance and spindle speed on the grinding force and torque obtained in the plane grinding of glass substrate edges, where the grinding conditions used were feed rate of 10,000 mm/min, depth of cut of 0.15 mm, and high-pressure coolant. Here, each grinding test was repeated three times and the presented grinding forces and torque were the average values of the three measurements. As seen from Figure 5a, grinding force ($F_x$) in the horizontal direction was almost constant while grinding force ($F_y$) in the vertical direction was trended to increase when increasing the grinding distance. As shown in Figure 5b, both grinding force ($F_x$) in the horizontal direction and grinding force ($F_y$) in the vertical direction were trended to decrease with the increase of grinding speed, while grinding torque was slightly increased with the increase of grinding speed, as shown in Figure 5c.

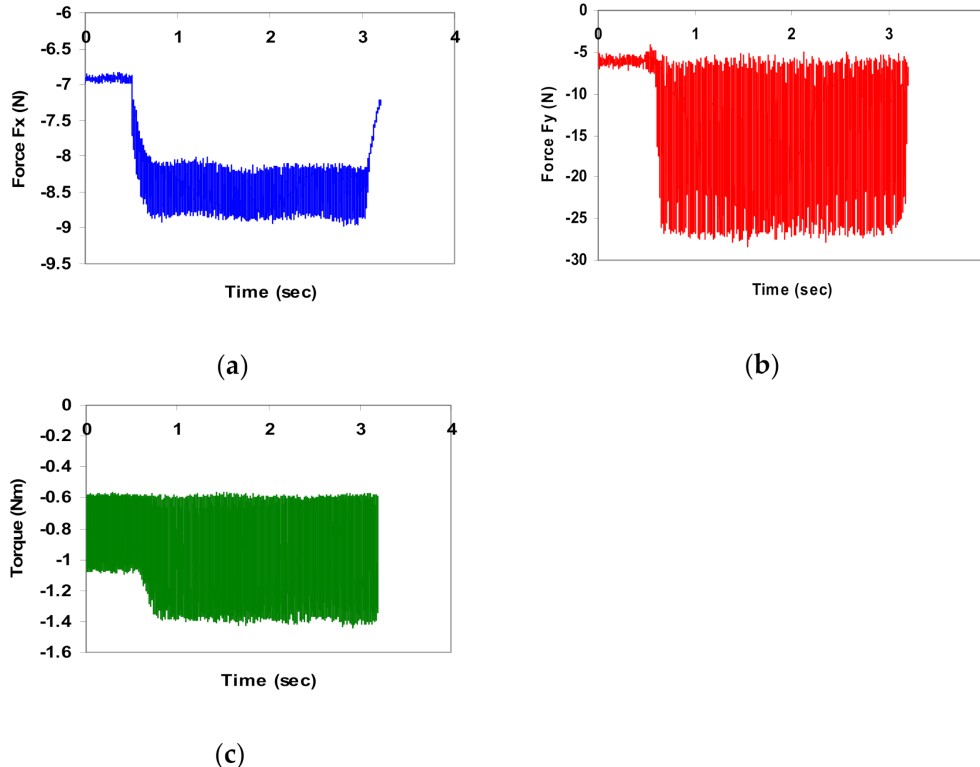

**Figure 4.** Grinding forces and spindle torque measured in the plane grinding of glass substrate edges: (**a**) grinding force, $F_x$; (**b**) grinding force, $F_y$; (**c**) grinding spindle torque.

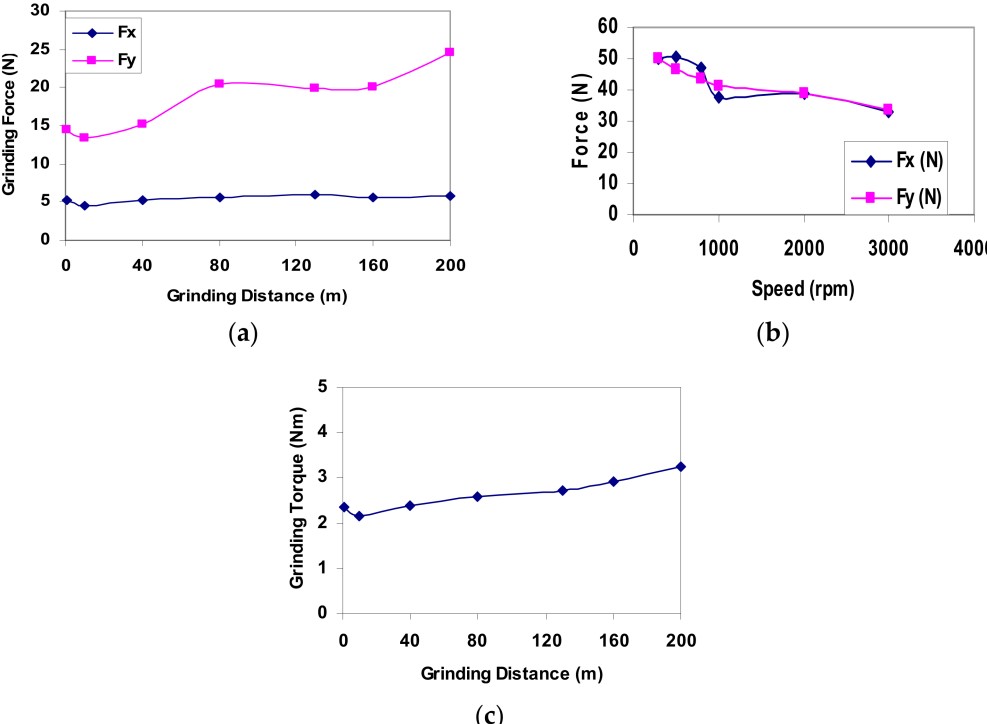

**Figure 5.** Effects of grinding distance and speed on grinding force and torque: (**a**) grinding force vs. grinding distance; (**b**) grinding force vs. spindle speed; (**c**) grinding torque vs. grinding distance.

### 4.2. Grinding Temperature

Grinding temperature was measured using a thermocouple embedded in glass substrate workpiece, where the grinding conditions used were grindings speed of 26.2 m/s (2000 rpm); feed rate of 7000 mm/min and 10,000 mm/min; depths of cuts of 0.1, 0.2, and 0.4 mm. As seen from Figure 6a, the grinding temperature was slightly increased with the increase of grinding depth when applying high-pressure coolant, while the temperature increase was much faster when a higher feed rate was used. However, when grinding under drying condition, the grinding temperature was increased extremely fast with the increase of grinding depth, as shown in Figure 6b. Compared the temperature rising obtained under the two conditions, the grinding temperature obtained under dry conditions was one order higher than that obtained when applying high-pressure coolant. Moreover, the grinding temperature reached over 1100 °C when grinding under dry conditions and with the grinding depth of 0.4 mm, of which the high grinding temperature may melt the grinding chips produced in the plane-grinding of glass substrate edges.%endparacol

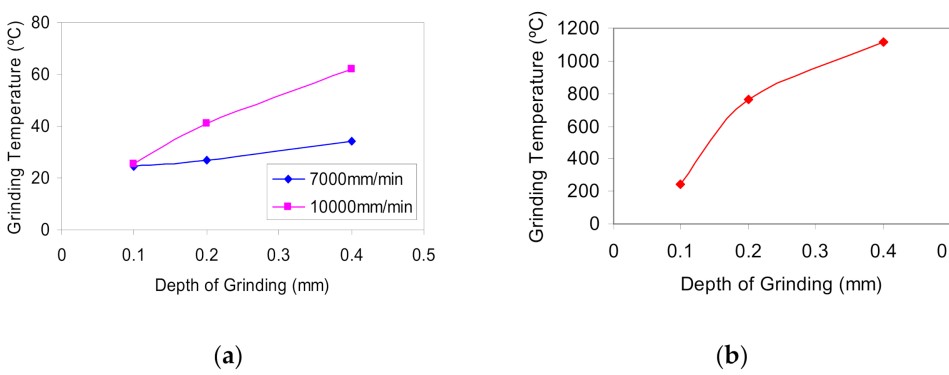

(**a**)                    (**b**)

**Figure 6.** Grinding temperature measured in plane grinding of glass substrate edges: (**a**) grinding under high-pressure coolant; (**b**) grinding under dry conditions.

### 4.3. Ground Surface Topography

Ground surface topography was examined using an optical microscope and SEM, as shown in Figure 7, where the grinding conditions used grinding speed of 26.2 m/s (2000 rpm), feed rate of 10,000 mm/min, depth of cut of 0.15 mm, and high-pressure coolant. Figure 7a is an overview of the ground glass edge topography, showing a very rough grinding surface generated. Figure 7b is a close view of the ground glass edge topography, showing that the ground surfaces were generated primarily by brittle fracture with a particle or chip adhered on the ground glass surface and the evidence of localized flow in the plane grinding of glass substrate edges. As seen from Figure 7c, a few small balls circled by redlines were found to be adhered to the ground glass surface, which is believed to be the melted or softened grinding chips/debris solidified by the high-pressure flush coolant under room temperature.

In fact, those small balls adhered on the ground glass edge as shown in Figure 7c were the high potential sources causing the glass substrate cracking in the subsequent integrated circuit (IC) printing process, where a big temperature fluctuation was experienced. It will be extreme costly to stop the production line for cleaning if the ground glass substrate is cracked and broken on the line. Therefore, it is very important to control the grinding chips/debris flowing direction during the grinding of glass substrate edges, so as to illuminate those melted or softened grinding chips/debris adhered to the ground glass edge and non-ground glass panel surface.

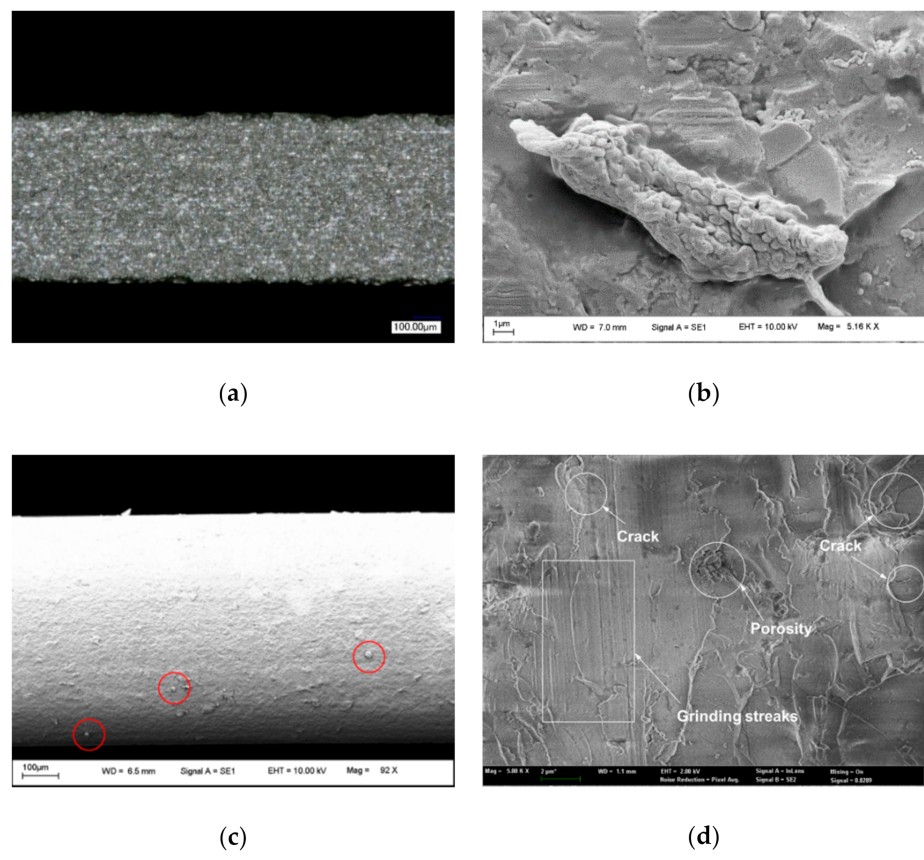

**Figure 7.** Ground edge surface topography and defects: (**a**) overview of ground edge; (**b**) close view of ground edge; (**c**) chips adhered on the ground edge; (**d**) defects on ground edge.

SEM observation of the defects produced on the ground glass substrate edge typography is shown in Figure 7d, where the marked white-line circles are cracks and porosity produced, and the marked white line rectangle is the grinding streaks that are the compression-induced abrasion marks [20]. Those cracks and porosity are the evidence of brittle mode grinding of glass substrate, and those parallel grinding streaks are also the evidence of ductile mode grinding behavior. Therefore, the grinding of glass substrate edges is behaved as brittle mode grinding under the above-mentioned experimental conditions. However, partial grinding of glass substrate edge was experienced under ductile mode grinding. This agrees with the theoretical analysis in Section 2 that the grinding of glass substrate edge was under brittle mode, and fractured surface was obtained when grinding the glass substrate using the above-mentioned conditions.

### 4.4. Diamond Wheel Wear

The diamond grinding wheel topography was examined before and after grinding using an optical microscope and SEM, as shown in Figures 8 and 9. A fresh diamond-grinding wheel is shown in Figure 8a, where the diamond grits were well protruded out from the resin bonder. Figure 8b shows the used diamond grinding wheel topography after grinding of 500 cycles (150 m), where some grooves with abrasive traces were found on the diamond wheel surface, which was parallel to the grinding direction, indicating the occurrence of a typical abrasion wear. These phenomena may be attributed to the soft resin binder of the diamond wheel being abraded by the display glass substrate.

The wear patterns of the used grinding wheel are shown in Figure 9. A protruded diamond grit is shown in Figure 9a on the used diamond wheel surface, where the grinding direction is from right to left. It was found that the soft binder in the front of the diamond grit was removed more than that behind the diamond grit, indicating that the diamond grit protruded freshly. Figure 9b shows a worn diamond grit on the used diamond wheel

surface. It is clearly indicated that the soft binder was removed equally surrounding the diamond grit and the diamond grit experienced the grit fracture during the grinding of glass substrate. Figure 9c shows a pocket on the used diamond wheel surface, indicating a pull-out of high protruding loosely held diamond grit.

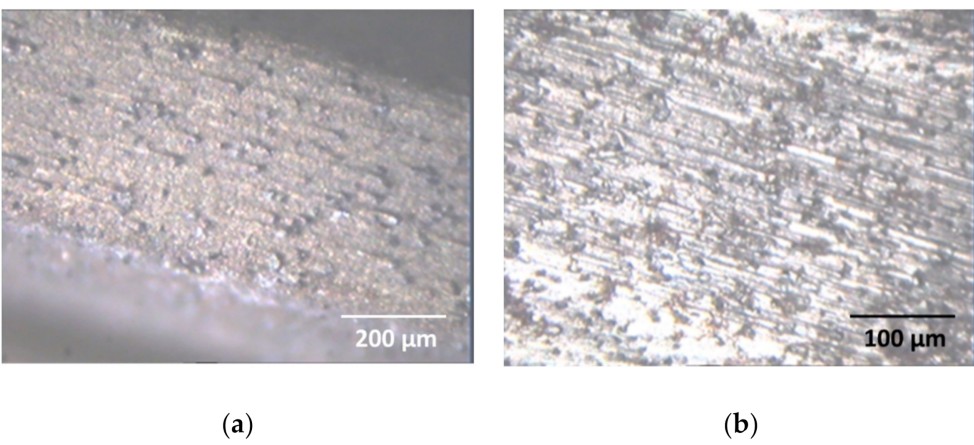

(a)                                                                (b)

**Figure 8.** Diamond grinding wheel surface topography before and after usage: (**a**) fresh grinding wheel; (**b**) used grinding wheel.

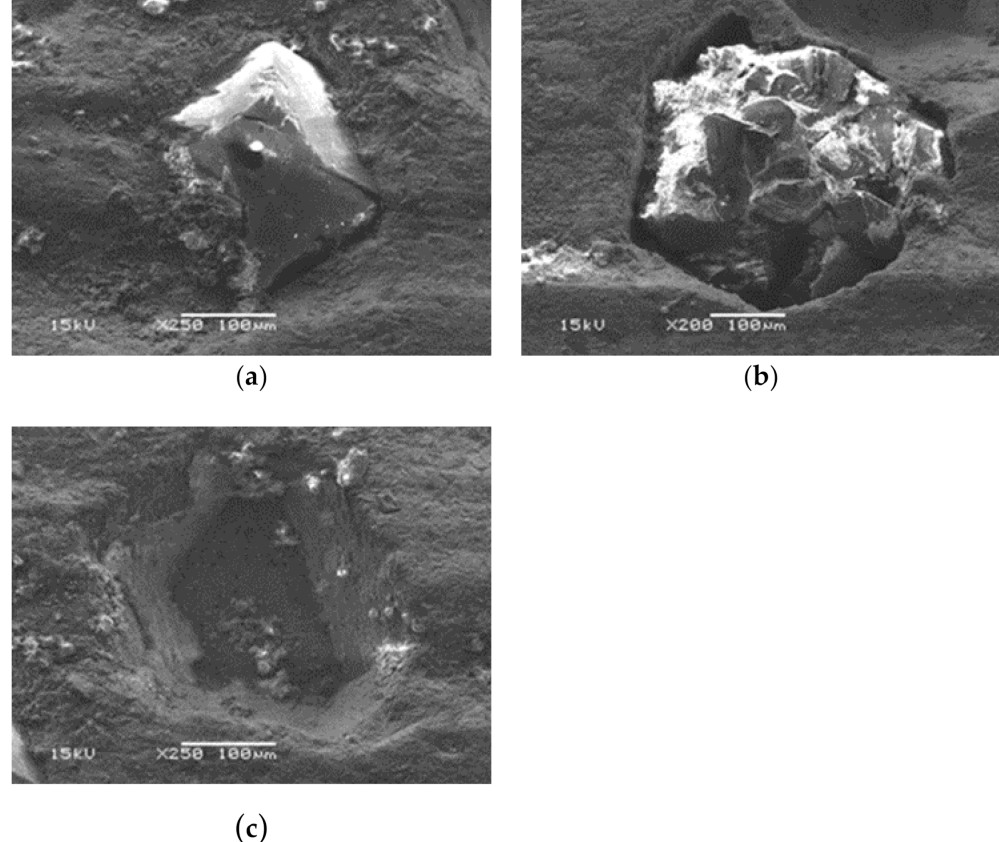

(c)

**Figure 9.** Diamond grinding wheel wear patterns: (**a**) a protruded diamond grit; (**b**) a worn diamond grit; (**c**) diamond grit pull out.

## 5. Discussion

In terms of plane-grinding glass substrate with a #500 resin-bound diamond wheel with a diameter $d_s$ of 250 mm and the diamond abrasive concentration rate of 75%, using feed rate $v_w$ of 10,000 mm/min, grindings speed of 26.2 m/s (2000 rpm), and depth of

cut $a_e$ of 0.15 mm, thus here $C$ is given by 40 and the semi-induced angle $\alpha$ is 60° [19]. Its maximum undeformed chip thickness for an individual grinding abrasive grit $h_{max}$ calculated using Equation (2) was 2.60 μm, which was much larger than the critical depth of cut $d_c$ of 40 nm for ductile mode grinding of glass substrates. Both the theoretical analysis and experimental results shown in Figure 7 clearly indicate that the edge grinding of glass substrates using the above-mentioned grinding conditions was under brittle mode grinding, while cracks were found on the ground surface of glass substrates. In the production, when grinding with a formed resin-bound diamond wheel having the same wheel diameter, grit size, and abrasive concentration rate but increasing the grinding wheel speed $v_s$ to 3000 rpm, and reducing depth of cut $a_e$ to 0.1 mm, while other grinding conditions remaining unchanged, we found the calculated maximum undeformed chip thickness for an individual grinding abrasive grit $h_{max}$ of 1.92 μm to be still greatly larger than the critical depth of cut $d_c$ of 40 nm for ductile mode grinding of glass, which again confirmed that fractured surfaces were generated in grinding of glass substrates with the formed diamond wheel due to brittle mode grinding, as shown in Figure 10a.

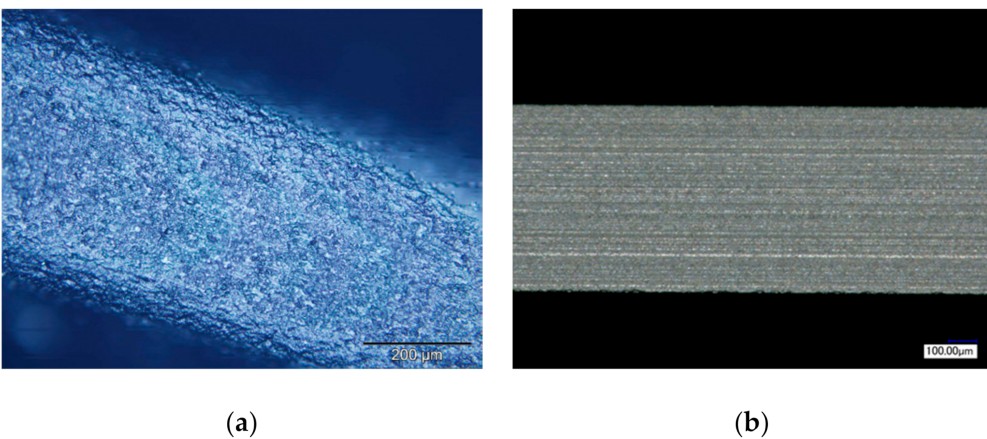

| (**a**) | (**b**) |
|---|---|

**Figure 10.** Ground surfaces of glass substrate edges: (**a**) fractured surface obtained under the spindle speed of 3000 rpm, feed rate of 7000 mm/min, and depth of cut of 0.1 mm; (**b**) smooth surface obtained under the spindle speed of 6000 rpm, feed rate of 5 mm/min, and depth of cut of 0.05 mm.

When further increasing the grinding speed $v_s$ to 78.5 m/s (6000 rpm), reducing depth of cut $a_e$ to 0.05 mm, and largely reducing feed rate to 5 mm/min for plane grinding, we found the calculated maximum undeformed chip thickness for an individual grinding abrasive grit $h_{max}$ of 25.5 nm to be smaller than the critical depth of cut $d_c$ of 40 nm for ductile mode grinding of glass substrates, which indicates that ductile mode grinding and smooth surfaces could be achieved under this grinding condition. Observation of the surface morphology of ground glass surfaces shown in Figure 10b is in agreement with the above-mentioned material removal mechanism, of which the ground surface exhibited a smooth appearance produced by ductile mode grinding of glass substrates. In addition, there were some grinding marks on the ground surface showing the evidence of localized flow, which were parallel to the grinding direction.

As shown in Figure 9b, some micro sharp edges were generated on the top of fractured diamond grits during grinding of glass substrates, playing the roles as micro cutting edges to perform the material removal [21,22]. Thus, partial grinding of glass substrate edge experienced ductile mode grinding behavior was attributed to and contributed by those micro cutting edges, which can be seen as a self-explanation of partial grinding performed under ductile mode, although the overall grinding behavior was under brittle mode.

Edge grinding is commonly used in the production of display glass panels to round the sharp edges and remove cutting defects of the glass edges. The strength of a display glass panel is actually a measure of the surface defects' size and number in the glass panel. The larger the defects, the weaker the glass panel will be [2]. Although ductile mode

grinding can be achieved under certain conditions, as shown in Figure 10b, the feed rate of 5 mm/min used is extremely slow. As a great penalty, the grinding process will be the bottleneck of the whole production and significantly low down the productivity line, which is definitely not acceptable by the industry. In fact, it is unavoidable to experience microcrack propagation and subsurface damages even after fine grinding of brittle materials including glass substrates [7,23–25]. Therefore, the polishing process is employed as a necessary subsequent manufacturing process to totally remove the grinding defects and subsurface damages for the production of display glass panels.

## 6. Conclusions

Theoretical analysis and experimental study on edge grinding of display glass substrate were conducted to investigate its grinding characteristics using a resin-bound diamond wheel under different grinding conditions. Some findings are summarized below:

- Grinding force ($F_y$) in the vertical direction obtained in the test conditions was much larger than grinding force ($F_x$) in the horizontal direction, causing a large compressive stress acting on the grinding glass edge.
- Grinding force ($F_x$) was almost constant, while grinding force ($F_y$) tended to increase when increasing the grinding distance up to 200 m. Grinding torque was slightly increased with the increase of grinding speed.
- Grinding temperature rising was slightly increased with the increase of grinding depth from 0.1 to 0.4 mm when applying high-pressure coolant, and it was increased much faster when a higher feed rate was used. Grinding temperature was measured as over 1100 °C under dry grinding.
- Ground surface topography analysis indicated that the grinding of glass substrate edge in the test conditions was performed under brittle mode machining, and fractured surface was obtained in the production.
- It was found that grinding of glass substrate edge was performed partially under ductile mode machining in the experimental conditions, which can be attributed to and contributed by those micro cutting edges generated by the fractured diamond grit on the grinding wheel surface.
- Ductile mode machining of glass substrate and smooth surface can be achieved under certain conditions when the critical undeformed chip thickness was less than 40 nm. However, the productivity for ductile mode machining conditions is extremely low, and thus subsequent polishing process is essential for display glass panel production.

**Author Contributions:** Conceptualization, K.L. and H.W.; methodology, K.L.; formal analysis, D.W.K.N. and R.H.; resources, R.H.; data curation, H.W.; writing—original draft preparation, D.W.K.N. and R.H.; writing—review and editing, K.L.; supervision, K.L. All authors have read and agreed to the published version of the manuscript.

**Funding:** This research received no external funding.

**Data Availability Statement:** Not applicable.

**Acknowledgments:** The authors would like to thank Ng Seow Tong and Shaw Kah Chuan Sean for help on conducting experiments.

**Conflicts of Interest:** The authors declare no conflict of interest.

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
