# Peer review of "Edge Grinding Characteristics of Display Glass Substrate"

_jmmp, doi:10.3390/jmmp5010020_

Round 1
Reviewer 1 Report
The subject of paper is scientifically interesting and fitted to the scope of Journal. Nevertheless, in order to accept, paper should be subjected to some revision. The detailed remarks are as follows:
- Introduction section should be modified and extended in order to improve readability. Avoid blocks of references, e.g. “ (...) to enhance their ductile mode cutting performance [8-12]. ”, as these do not emphasize the particular aspects from the cited papers. Particularly, when citations are made in reference to specific technical aspects, single/double, e.g. [1, 2] references are encouraged. It is strongly suggested that the references need to make in-depth comments on the content of the cited papers; avoid generic comments. Mention/comment the relevance of the cited paper and especially the research gap associated to it.
- The literature survey seems to be limited. In order to reflect the actual state-of-the-art some recent works concerning precise cutting should be added. Among these, the following works should be discussed:
-Surface texture formation in precision machining of direct laser deposited tungsten carbide. Advances in Manufacturing (2017) Volume 5, Issue 3, pp 251–260.
-The study on minimum uncut chip thickness and cutting forces during laser-assisted turning of WC/NiCr clad layers. The International Journal of Advanced Manufacturing Technology (2017), doi:10.1007/s00170-017-0035-5.
-Prediction of cutting forces during micro end milling considering chip thickness accumulation, International Journal of Machine Tools and Manufacture (2019), 103466, https://doi.org/10.1016/j.ijmachtools.2019.103466
- At the end of Introduction please cleraly demonstrate what is the novelty of the proposed work towards the existing ones in this scope.
- In experimental details please define the directions of the measured forces
- What kind of valuable/scientific information to analysis provide charts depicted in Fig. 4? Are they really needed? Moreover, it seems that before the initiation of cutting, the measured noise of force and torque signals was different than 0, why? It could affect the accuracy of measurements.
- What kind of statistical measure of force and torque is presented in Fig. 5? Was it the mean arithmetic value, RMS, or something else?
Author Response
Thanks for the reviewer's time and comments. We have modified our paper according to the reviewer's comments marked in RED.
- The introduction section has been updated accordingly to the comments marked in RED.
- One of the papers listed in the reviewer comment has been added in the references, while the other two papers are not relevant to our paper.
- Fig. 1 has been updated with measured force directions.
- Regarding Fig. 4, more explanation has been added in as "As the four-component Kistler dynamometer was mounted to the grinding spindle in the horizontal direction, which usually is mounted on a vertical milling spindle to measure machining forces and torque, the mass of the dynamometer together with the grinding wheel acts as a pre-loading. Thereafter, all the measured forces Fx and Fy have an initial value and the measured torque has an off-set value before the actual grinding in response to the pre-loading as shown in Figure 4. "
- In Fig. 5 presented is the average grinding forces and torque as "Here, each grinding test was repeated 3 times and the presented grinding forces and torque were the average values of the three measurements."
Reviewer 2 Report
Dear authors,
I have a few comments on your article:
- On line 111 you have mentioned the measurement of surface roughness using a profilometer but nowhere in the article do you have the results, please add these results.
- What causes smaller values of forces Fx and Fy in the first half second of the force measurement on Fig. 4 (a) and Fig. 4 (b). please indicate whether this is caused by start-up of the grinding tool or another reason.
- Please add a magnification to the individual pictures in Fig. 7, which is not legible somewhere.
- In chapter 4.4 (line 189), which describes the wear of the grinding wheel, you are missing the time or number of cycles after which the damage you documented occurred.
- The conclusions are very general, try to elaborate the achieved results more, eg. with the indication of specific grinding parameters, etc.
- In the used references there are 6 citations of co-author Kui.L. Out of the total number of 26 literary sources, this author is mentioned in more than 20%. Please reduce the number of resources by this author.
Author Response
Thanks for the reviewer's time and comments. We have modified our paper according to the reviewer's comments marked in RED.
- On line 111, the sentence “Surface roughness was measured using a stylus surface profilometer.” has been removed as the surface roughness is not reported and discussed in this paper.
- Regarding Fig. 4, more explanation has been added in as "As the four-component Kistler dynamometer was mounted to the grinding spindle in the horizontal direction, which usually is mounted on a vertical milling spindle to measure machining forces and torque, the mass of the dynamometer together with the grinding wheel acts as a pre-loading. Thereafter, all the measured forces Fx and Fy have an initial value and the measured torque has an off-set value before the actual grinding in response to the pre-loading as shown in Figure 4. "
- A magnification scale has been added in Fig. 8.
- In Chapter 4.4, modified the sentence as "Figure 8 (b) shows the used diamond grinding wheel topography after grinding of 500 cycles (150 m)," to indicate the number of cycles.
- The Conclusion section has been modified by adding-in the specific grinding parameters.
- Two references have been removed according to the comments as: " 16. Liu, K.; Wang, H.; Zhang, X.Q. Modelling of ductile mode cutting. In Ductile Mode Cutting of Brittle Materials, Springer Nature: Singapore, 2020; pp. 55-73. and 20. Liu, K.; Wang, H.; Zhang, X.Q. Ductile mode cutting characteristics. In Ductile Mode Cutting of Brittle Materials, Springer Nature: Singapore, 2020; pp. 39-53.".
Reviewer 3 Report
- The paper addresses a topic of current interest (processing glass display substrates) and in accordance with the profile of the journal.
- The authors used modern research equipment and methods. No new or improved solutions for equipment or methods for researching the investigated manufacturing process have been proposed.
- A cutting scheme would have been useful for a better understanding of how the edges are ground and how the components of the cutting force are positioned.
- Table 2 does not mention the melting temperature of the glass. At temperatures exceeding 1000 oC (for example, at the temperatures indicated in figure 6, b), can't the glass melt?
- In the case of the photographs in figures 8 a and b, it would be useful to highlight the magnification scale in order to be more correctly noticed and interpreted the differences between the two photographs.
- It seems more appropriate to highlight the influence of cutting speed (m/min or m/s) on the magnitude of the cutting force than to consider the influence of the rotational speed of the abrasive tool (r/min) on the magnitude of the force since the cutting speed is the one that determines the way of removing material from the workpiece and not the rotation speed of the tool.
- It is usually appreciated that the concept of "machining" (processing with the removal of material from the workpiece) includes that of "grinding" and, in this case, the wording "When machining and/or grinding of brittle materials" is not proper.
- Is it possible that "a resign bound diamond grinding wheel" means "a resin-bound diamond grinding wheel"?
- Authors may pay more attention to editing the paper. Thus, confusing forms may be encountered in the paper or the breaking of longer sentences was performed in an inappropriate manner (e.g., "Because there is a large reduction in device size and mass while achieving high-quality display.", " While there is a demand for thinner glass substrates for display because of the market-driven of lighter devices and material-saving. ”, “ While the maximum undeformed chip thickness for an individual grinding abrasive grit, hmax, can be calculated by [19 ]: ”, etc.
In the case of the formulation „Substituted the values of glass’s elastic modulus, hardness, and critical fracture toughness into Eq. (1), ”, probably more correct could be“ Substituting the values of glass’s elastic modulus, hardness, and critical fracture toughness into Eq. (1),".
Between the number and the units of measurement, it is necessary to insert a blank space. For example, write "250 mm", instead of "250 mm".
It is proposed to replace the wording "on work material's properties" with "on workpiece material's properties", given the widespread use of the concept "workpiece" to designate the object affected by the machining process.
Author Response
Thanks for the reviewer's time and comments. We have modified our paper according to the reviewer's comments marked in RED.
- All typing errors have been corrected accordingly.
- All the technical terms suggested have been updated accordingly.
- The Introduction section has been rewritten more clearly.
- Fig. 1 has been updated with measured force directions.
- A magnification scale has been added in Fig. 8.
- Table 2 has been updated by adding in "Strain point (°C), Annealing Point (°C) and Softening Point (°C)" of the workpiece material. As the grinding feed rate is very fast, even the grinding temperature rising up to 1000°C,, our observation is there is no melting of glass edge occurred.
Round 2
Reviewer 1 Report
Authors have carefully revised their manuscript according to the reviewer's remarks. Therefore, in a current form it can be accepted for a publication.